# New Time-Related Insights into an Old Laboratory Parameter: Early CRP Discovered by *IBM Watson Trauma Pathway Explorer©* as a Predictor for Sepsis in Polytrauma Patients

**DOI:** 10.3390/jcm10235470

**Published:** 2021-11-23

**Authors:** Ladislav Mica, Hans-Christoph Pape, Philipp Niggli, Jindřich Vomela, Cédric Niggli

**Affiliations:** 1Department of Trauma Surgery, University Hospital Zurich, 8091 Zurich, Switzerland; Hans-Christoph.Pape@usz.ch (H.-C.P.); cedric.niggli@swissonline.ch (C.N.); 2Department of Mathematics, ETH Zurich, 8092 Zurich, Switzerland; philipp.niggli@swissonline.ch; 3Division of Medical Sciences in Sports, Masaryks University, 62500 Brno, Czech Republic; jvomela@mail.muni.cz

**Keywords:** WATSON Trauma Pathway Explorer, artificial intelligence, CRP, prediction, sepsis, polytrauma

## Abstract

The University Hospital Zurich together with IBM^®^ invented an outcome prediction tool based on the IBM Watson technology, the *Watson Trauma Pathway Explorer^®^*. This tool is an artificial intelligence to predict three outcome scenarios in polytrauma patients: the Systemic Inflammatory Response Syndrome (SIRS) and sepsis within 21 days as well as death within 72 h. The knowledge of a patient’s future under standardized trauma treatment might be of utmost importance. Here, new time-related insights on the C-reactive protein (CRP) and sepsis are presented. Meanwhile, the validated *IBM Watson Trauma Pathway Explorer^®^* offers a time-related insight into the most frequent laboratory parameters. In total, 3653 patients were included in the databank used by the application, and ongoing admissions are constantly implemented. The patients were grouped according to sepsis, and the CRP was analyzed according to the point of time at which the value was acquired (1, 2, 3, 4, 6, 8, 12, 24, and 48 h and 3, 4, 5, 7, 10, 14, and 21 days). The differences were analyzed using the Mann–Whitney U-Test; binary logistic regression was used to determine the dependency of prediction, and the Closest Top-left Threshold Method presented time-specific thresholds at which CRP is predictive for sepsis. The data were considered as significant at *p* < 0.05, all analyses were performed in R. The differences in the CRP value of the non-sepsis and sepsis groups are starting to be significant between 6 and 8 h (*p* < 0.05) after admission inclusive of post hoc analysis, and the binary logistic regression depicts a similar picture. The level of significance is reached between 6 and 8 h (*p* < 0.05) after admission. The knowledge of the outcome reflected by the CRP in polytrauma patients improves the surgeon’s tactical position to indicate operations to reduce antigenic load and avoid an infectious adverse outcome.

## 1. Introduction

Overwhelming systemic inflammatory reactions, both polytrauma-triggered and/or microbial triggered, contribute to an adverse outcome not only physiologically but also lowering the operative success, especially in the field of trauma [1,2]. The damage control concept in the treatment of polytrauma patients takes account to avoid infectious and inflammatory complications but cannot be monitored in any way. This study group, in cooperation with IBM, developed a predictive tool, more an artificial intelligence (AI), the *IBM Watson Trauma Pathway Explorer©*, to predict the outcome of critically injured patients, as described previously [3,4]. This dynamic online application includes prognostic parameters that allow an estimation of an adverse outcome: early death within 72 h since admission and systemic inflammation as well as sepsis within 21 days since admission. However, it also depicts the timeline of frequently used laboratory parameters in clinical daily life, such as the C-reactive protein (CRP). The difference in the CRP was apparent between the patients suffering sepsis and those who did not within hours after admission. At this point, the hypothesis arose that the reduction of CRP by surgical or medical interventions at a very early stage after the admission of a critically injured patient could improve the outcome. The interactions of CRP are widespread, including the activation of the complementary system (C3b), metabolic enhancement (activation of glucose-6-phosphatase), opsonization of polysaccharides lecithin, and nucleic acid improving immunological bacterial and debris clearance in a critically injured patient [5,6,7]. The role of the CRP might be summarized to an immunological, vascular, pro-coagulative, and pro-metabolic function changing trauma surgeons’ view on this small molecule. As demonstrated in this study, the role of the CRP could be rethought especially when observed also in the dimension of time. Certainly, a detailed time-dependent CRP analysis had to follow.

## 2. Methods

### 2.1. Ethical Statement

The study was conducted according to the guidelines for good clinical practice and the Helsinki guidelines. The research was based on the TRIPOD Statement, which is a guideline for multivariable prediction models [8]. The analysis of patient records has been approved by the ethical committee of the University Hospital Zurich and the government of Zurich upon the development of the database (Nr. StV: 1-2008) and reapproved to develop the *Watson Trauma Pathway Explorer©* (BASEC: 2021-00391).

### 2.2. Patient Sample, Inclusion and Exclusion Criteria

Eligibility criteria for the patients were age ≥ 16 years and ISS ≥ 16. Patients with an ISS equal to or above 16 are considered to have an injury or several injuries that correspond to a polytrauma. Patients were admitted to the trauma bay primarily, and only those with complete datasets were included. Patients referred from another hospital were excluded, as well as non-survivors on the scene. The sample was divided into a group without sepsis and a second group suffering sepsis during the observational period of 21 days. In total, 3653 patients were used for *Watson Trauma Pathway Explorer*©. The database was established on 01.08.1996 with ongoing patient data collection. The CRP values were measured at different points of time in a daily routine as follows: 1, 2, 3, 4, 6, 8, 12, 24, and 48 h and 3, 4, 5, 7, 10, 14, and 21 days after admission into the trauma bay of the University Hospital Zurich.

### 2.3. Definition of Sepsis

The worst parameters of leukocyte count, respiratory rate, heart rate, and temperature were taken to determine the SIRS score each day [9]. SIRS was measured during the first 30 days after admission or as long as the patients were hospitalized. Sepsis was defined as a SIRS score ≥ 2 with an infectious focus. The sepsis had to occur at any time during the observational period of 21 days.

### 2.4. CRP Measurement

The CRP was measured in the *Institut für Klinische Chemie* at the University Hospital Zurich by a standardized latex-enhanced immune turbidimetry [10]. The same analyzing method for all patients at all time points was applied. The used dimension is mg/L. Taxonomy: 9606 [NCBI], NX_P02741.

### 2.5. Statistical Analysis

The baseline characteristics of the patients’ sample were described through means with standard deviation (SD) for numerical variables, medians with interquartile ranges (IQR) for ordinal data, and percentages for binary variables. An unpaired *t*-test for numerical variables and Mood’s median test for ordinal variables assessed the differences between these groups.

Data were tested for normality with a Q-Q-Plot. The Mann–Whitney U-Test was used to determine the significance between the groups because the data were not normally distributed and the variance was not equal. Data were considered as significant if *p* < 0.05. Binary logistic regression was performed to determine independent predictive ability. The threshold values for CRP at different time points were determined by the Closest Top-left Threshold Method. This method calculates the threshold point that is closest to the top-left of the ROC plot of each CRP time point. Statistics were performed with R-4.0.2.

## 3. Results

### 3.1. Characteristics of the Patient Sample

In total, 3653 patients were included. About 75% were male in both study groups (Table 1). The trauma mechanism was mostly blunt. The ISS was significantly higher in the sepsis group (30; IQR 25–41 vs. 25; IQR 17–34, *p* < 0.001) (Table 1). Likewise, the APACHE II score was significantly higher in the sepsis group compared to non-septic patients (17; IQR 11–21 vs. 13; IQR 6–21, *p* < 0.001) (Table 1).

### 3.2. Significant Differences in the CRP between the Sepsis Groups

The Q-Q plots have no normality shown. The data were tested with the Mann–Whitney U-test and started to show significant differences between 6 and 8 h after the admission of the severely injured patient. The data remained significant 8 h after admission and over the whole observational period (*p* < 0.05) (Figure 1).

### 3.3. CRP as an Independent Predictor for Sepsis

Binary logistic regression was performed depicting a similar picture of significance as in the Mann–Whitney analysis. The data started to be significant between 6 and 8 h after admission and persisted to be significant over the complete observational period (*p* < 0.05) (Figure 2).

### 3.4. The Predictive Quality Is Not Balanced, ROC Is Far Too Low

The analysis of the patient sample by ROC revealed no satisfying values of the area under the ROC (AUROC). The AUROC was always <0.800. In detail, AUROC 0.643 after 6 h and AUROC 0.583 after 8 h.

### 3.5. Threshold Values as Orientation Points

The Closest Top-left Threshold Method was applied to test the patient sample. The result was a bell-shaped curve with a peak after three days (Figure 3). This diagram reflects the period of CRP values from 1 h after admission to 21 days after admission; interestingly, the threshold for sepsis at 8 h is CRP 9.9. The threshold values reached their maximum after 3 days at CRP 132.5 (Figure 3).

## 4. Discussion

The presented analysis of CRP in multiply injured patients with and without sepsis confirms the theory of SIRS (Systemic Inflammatory Response Syndrome) and CARS (Compensatory Anti-Inflammatory Response Syndrome) in a completely new fashion that was made possible by the IBM artificial intelligence, the *Watson Trauma Pathway Explorer©* [3,4,11].

The CRP is an acute-phase protein that depicts directly the systemic inflammatory state of a patient: here, in this case, the multiply injured patient. The inflammation comes from the trauma load as an early reaction to contamination, cell debris, and blood loss. A CRP increase and an inflammatory state are always observed in a polytrauma patient with an according injury pattern [12]. The higher the initial contamination and trauma load, the higher the CRP value might rise. During the time course of polytrauma management surgery and definitive care, the CRP level might variate, and at some point in time, the diagnosis of sepsis might be set [11,12,13]. Considering the whole situation in the time dimension and the SIRS/CARS theory, it could be imaginable that SIRS might be individually limited in its duration and severity, leading to an inflammatory burnout overtaken by a CARS-like situation as the dominating immunological reaction [11]. However, both SIRS and CARS are probably paralleled reactions, and what we see is only the dominant state of both [11]. Considering the patient sample as ISS-normalized sets the CRP values on the same level and makes the preview possible whether the CRP is too high for the ISS situation or not. This fact can be used between six and eight hours after admission, as shown in Figure 1 and Figure 2 for the first time. As shown by the threshold values, the CRP value at 8 h is quite low (Figure 3), but it is an independent predictor for the development of sepsis (Figure 1 and Figure 2). This is a very early point of time in the treatment of a polytrauma patient to be able to determine possible septic complications in a polytrauma setting. Certainly, the thresholds are rising during the time and might be used only with an according time course. The poor predictive quality (AUROC) might be seen from a multivariate perspective in the sense of lacking sepsis cases or the multivariate etiology of CRP increase in a polytrauma patient. The multivariate hypothesis seems to be rational when considering the ICU treatment procedures and operative interventions, leading to a CRP increase.

The *IBM Watson Trauma Pathway Explorer©* led the explorers to this study design and the statistical confirmation of the initially assumed theory. The given time points between six and eight hours to assess a potential septic time course is very early and might be used to guide the trauma surgeon in decision finding, monitoring surgical success, and planning of second-look procedures. The provided threshold values should be used with critical precautions. No weight or BMI (Body Mass Index) normalization was undertaken, and hepatopathology and system senescence were not taken into account. Additionally, routine-based drug administration, malnutrition, and katabolic nutritional states in severely injured patients were also not considered in this study as possible bias factors. No statement can be made about the number of intubated and ventilated patients on arrival. However, this could be relevant, as it may affect the risk of developing sepsis due to respiratory infections. Finally, minor changes in treatment recommendations over the last decade may have led to changes in CRP values after surgery.

## 5. Conclusions

The provided time course of the CRP values in polytrauma patients depicts very early the CRP as an independent predictor for septic complications. The CRP value might be used very early on as a marker for the success of initial surgical damage control intervention, even further on during the complete observational period of 21 days. Furthermore, early disease recognition could lower the mortality due to septic shock and lethal infectious courses by planning interventions and preemptive drug administration. A prospective randomized study design has to evaluate whether the findings are in concordance with the clinical reality of polytrauma patients.

## Figures and Tables

**Figure 1 jcm-10-05470-f001:**
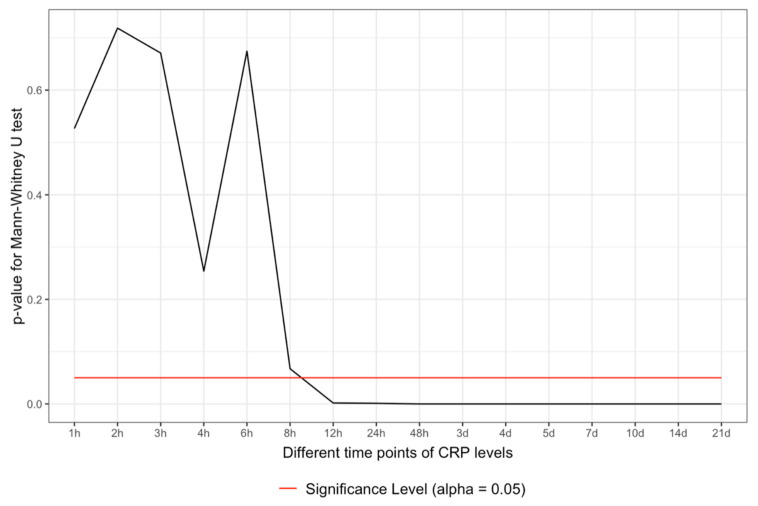
Mann–Whitney U-test between the sepsis and non-sepsis group according to the points of time. As indicated, the differences start to be significant between 6 and 8 h after admission (sepsis vs. no sepsis).

**Figure 2 jcm-10-05470-f002:**
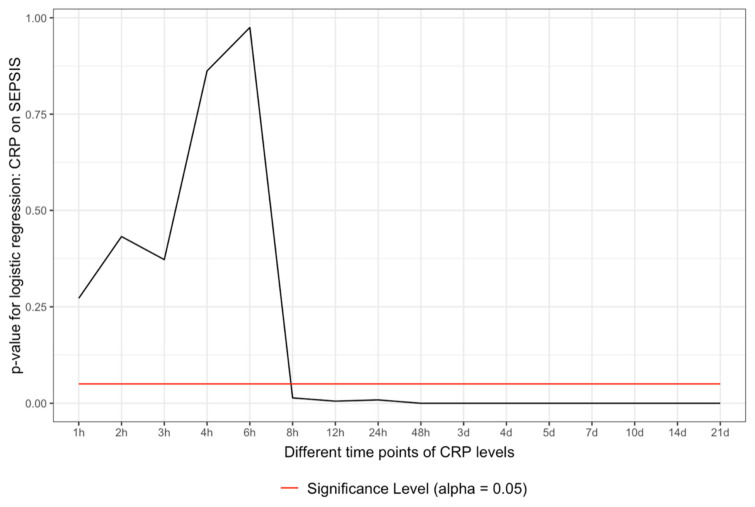
CRP is an independent predictor of sepsis. Binary logistic regression of the CRP values and the two groups (sepsis vs. no sepsis). In addition, here, the values are significant between 6 and 8 h.

**Figure 3 jcm-10-05470-f003:**
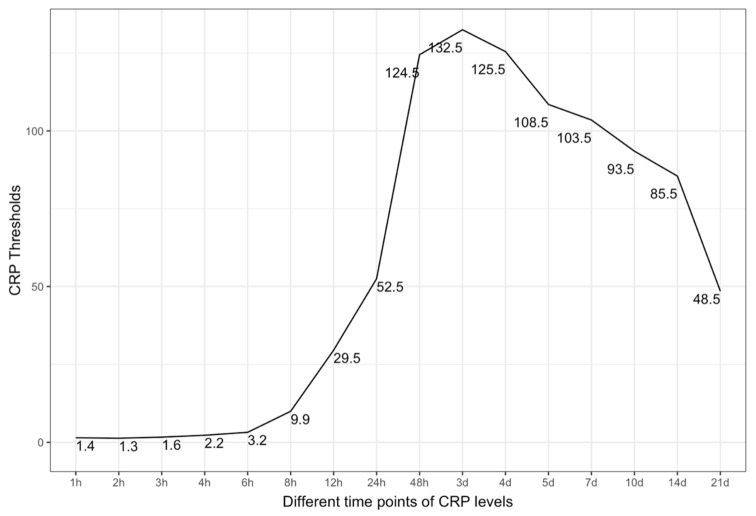
Closest Top-left Threshold Method of the patient sample. Shown are the threshold values of the CRP levels, which are predictive for sepsis at a given time point (over the whole observational period of 21 days).

**Table 1 jcm-10-05470-t001:** Descriptive statistics of the patient sample. Shown are the CRP values for 6, 8, 12, and 24 h only.

	Patient Sample *N* = 3653	Sepsis *N* = 547	No Sepsis *N* = 3106	*p*-Value
Age (mean, SD)	45.8 ± 20.2	42.8 ± 18.1	46.3 ± 20.5	0.0002
Male	73.4%; *N* = 2681	78.6%; *N* = 430	72.4%; *N* = 2251	-
Early death within 72 h	19.3%; *N* = 708	14.6%; *N* = 8	22.5%; *N* = 700	-
Blunt trauma	91.3%; *N* = 3336	94.7%; *N* = 518	90.7%; *N* = 2818	-
Head injury	38.3%; *N* = 1400	44.8%; *N* = 245	37.2%; *N* = 1155	-
BMI at admission (mean, SD)	25 ± 4.4	25.9 ± 4.4	24.8 ± 4.3	<0.001
ISS (median, IQR)	25 (17–34)	30 (25–41)	25 (17–34)	<0.001
NISS (median, IQR)	34 (25–50)	41 (33–50)	34 (24–48)	<0.001
Temperature at admission (mean, SD)	35.5 ± 1.7	35.4 ± 1.7	35.6 ± 1.7	0.131
GCS at admission (median, IQR)	10 (3–15)	3 (3–14)	11 (3–15)	<0.001
pH at admission (mean, SD)	7.31 ± 0.13	7.29 ± 0.15	7.32 ± 0.13	0.006
Lactate at admission (mean, SD)	2.94 ± 2.53	2.94 ± 2.27	2.94 ± 2.58	0.943
Hemoglobin at admission (mean, SD)	11.4 ± 4	11 ± 2.8	11.5 ± 4.2	0.005
Quick at admission (median, IQR)	84 (65–97)	80 (61–92)	85 (66–98)	0.1257
Systolic blood pressure at admission (mean, SD)	130.7 ± 27.6	128.5 ± 27.7	131.2 ± 27.5	0.0715
APACHE II at admission (median, IQR)	14 (7–21)	17 (11–21)	13 (6–21)	<0.001
CRP at 6 h (mean, SD)	12.3 ± 29.6	11.3 ± 24.4	12.6 ± 31.5	0.049
CRP at 8 h (mean, SD)	19.7 ± 33.1	41.23 ± 60.74	15.5 ± 22.8	<0.001
CRP at 12 h (mean, SD)	37.3 ± 40.7	52.1 ± 55.6	34.3 ± 36.4	<0.001
CRP at 24 h (mean, SD)	71.1 ± 60.6	80.9 ± 68	68. ± 58.4	<0.001

## Data Availability

All data are available upon reasonable request. None of these data are available for a broad public. All data are stored in the clinical information system of the University Hospital Zurich.

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
