# Peer review of "New Time-Related Insights into an Old Laboratory Parameter: Early CRP Discovered by IBM Watson Trauma Pathway Explorer© as a Predictor for Sepsis in Polytrauma Patients"

_jcm, 2021, doi:10.3390/jcm10235470_

Round 1

Reviewer 1 Report

Dear authors, the following comments should give an overview why I came to this decision.

Language:

  • Both the spelling mistakes and the sentence structure creates the impression that the article has not been finished.

Introduction:

  • Spelling mistakes for example “Over helmimg”; “ patients suffering a sepsis a(nd) those..”; “polysaccharidies”
  • Syntax: “This study group developed in cooperation with IBM a predictive tool” -> maybe better “ this study group, in cooperation with IBM developed a predictive tool”; “…when some very early parameters are inputed” -> please choose a different word or sentence if possible, also define what is very early if possible

Materials and methods

  • Patient sample, Inclusion and Exclusion Criteria:
    1. “Totally 3953 patients were used …” -> why is it 3653 in the following data?
    2. In what period of time the patients were admitted and treated in your hospital (Years)? Regarding to your inclusion criteria it needs to be a longer period? Have there been any changes to the treatment recommendations or SOP´s in your clinic during this time? If so, this may be mentioned as a limitation later on.
  • CRP Measurement
    1. Did you use the same analysing method for all patients at all time points?

Results

  • As I mentioned before, in this section there are 3653 Patients included while in material and methods there are 3953 Patients, mistake in writing or mistake in data?
    1. “Tab.1.: Is it possible to add the rate of intubated and ventilated patients on arrival? This could be relevant because it could have an impact on the risk of developing sepsis based on respiratory infections. If lower GCS is mainly due to the higher rate of TBI this could also influence the outcome.
    2. 1. “CRP at different time points” in my opinion is wrong because you are showing time points on this axis and not CRP levels, so it should be “different time points of CRP Analysis/Levels”; same is for Fig 2 and 3
    3. 4. “ The analysis … revealed not (no?) satisfying values

Author contribution: Missing

Funding: Missing

Institutional Review Board Statement: Missing

Informed Consent Statement: Missing

Data Availability Statement: Missing

Acknowledgements: Missing

As mentioned before the intention of this study as well as the results are really interesting and further research is needed in this field. But by considering the above-mentioned aspects this article should be revised to improve the overall quality.

Author Response

Thank you very much for your review, which is really appreciated. Unfortunately, we did not submit the very last version to the journal, but the second last, which is why two numbers were not correct and the spelling was not yet perfect. The current version is correct. Please see the attachment for the comments.

Reviewer 2 Report

The paper is well written and focused on a topic of clinical relevance.

The novelty is represented by the cut-off of CRP which may timely identify the trauma patient who's developing septic complications.

My concerns are:

-Do the authors think that a subgroup analysis among patients who underwent damage control for hollow viscus injuries (isolated or in association with other conditions) in comparison to others who underwent DCS only for hemorrhagic conditions may reveal any differences in the time of increasing CRP if sepsis is developing, or not?

-Have you considered polytrauma patients intended as multisystem trauma or as severe trauma? Do you think that the systemic response and CRP may have different trends in case of a multisystem trauma in comparison to a severe one, or not. Could you better specify the type of patients you enrolled?

Author Response

Thank you very much for your review, which is really appreciated. Please see the attachment for the comments.
